# H_2_ Plasma and PMA Effects on PEALD-Al_2_O_3_ Films with Different O_2_ Plasma Exposure Times for CIS Passivation Layers

**DOI:** 10.3390/nano13040731

**Published:** 2023-02-14

**Authors:** Jehyun An, Kyeongkeun Choi, Jongseo Park, Bohyeon Kang, Hyunseo You, Sungmin Ahn, Rockhyun Baek

**Affiliations:** 1Department of Electrical Engineering, Pohang University of Science and Technology (POSTECH), Pohang 37673, Republic of Korea; 2National Institute for Nanomaterials Technology (NINT), Pohang University of Science and Technology (POSTECH), Pohang 37673, Republic of Korea

**Keywords:** high-k gate dielectric, Al_2_O_3_, H_2_ plasma treatment, interface trap, plasma-enhanced atomic layer deposition

## Abstract

In this study, the electrical properties of Al_2_O_3_ film were analyzed and optimized to improve the properties of the passivation layer of CMOS image sensors (CISs). During Al_2_O_3_ deposition processing, the O_2_ plasma exposure time was adjusted, and H_2_ plasma treatment as well as post-metallization annealing (PMA) were performed as posttreatments. The flat-band voltage (V_fb_) was significantly shifted (ΔV_fb_ = 2.54 V) in the case of the Al_2_O_3_ film with a shorter O_2_ plasma exposure time; however, with a longer O_2_ plasma exposure time, V_fb_ was slightly shifted (ΔV_fb_ = 0.61 V) owing to the reduction in the carbon impurity content. Additionally, the as-deposited Al_2_O_3_ sample with a shorter O_2_ plasma exposure time had a larger number of interface traps (interface trap density, D_it_ = 8.98 × 10^13^ eV^−1^·cm^−2^). However, D_it_ was reduced to 1.12 × 10^12^ eV^−1^·cm^−2^ by increasing the O_2_ plasma exposure time and further reduced after PMA. Consequently, we fabricated an Al_2_O_3_ film suitable for application as a CIS passivation layer with a reduced number of interface traps. However, the Al_2_O_3_ film with increased O_2_ plasma exposure time deteriorated owing to plasma damage after H_2_ plasma treatment, which is a method of reducing carbon impurity content. This deterioration was validated using the C–V hump and breakdown characteristics.

## 1. Introduction

Recently, the importance of CMOS image sensor (CIS) technology has rapidly increased owing to its relevance in mobile products and autonomous driving. As electronic products become ever-smaller in size, smaller CIS devices are also required. Therefore, CIS devices must be scaled, similar to other semiconductor devices. The pixel size of the CIS image sensor has been rapidly scaled, limiting the number of photons entering the pixel. In addition, as a result of scaling, light reflection occurred, causing light loss and cross-talk issues [1]. Therefore, a backside illumination-type CIS device that illuminates the rear side of the device was developed [2]. However, the backside illumination structure is adversely affected by dark currents and noise. Hence, in order to decrease dark current and increase quantum efficiency, research on the development of high-k materials for application as a CIS passivation dielectric layer is necessary.

Al_2_O_3_, which is a high-k dielectric material, has a wide energy bandgap and high thermal stability; therefore, it is suitable for application as a passivation dielectric film for CIS [3,4]. In addition, unlike other dielectric films, Al_2_O_3_ has negative fixed charges and shows excellent passivation characteristics [5]. In most semiconductor devices such as complementary metal oxide semiconductors (CMOSs), fixed charges act as defects [6]. Thus, many studies have been conducted to control these negative fixed charges [7]. However, in the CIS device, a passivation dielectric layer is required to contain high fixed charges for field effect passivation. Therefore, Al_2_O_3_ is a suitable dielectric material as a passivation layer of CIS. However, a dielectric film with fewer impurities is required for fabricating a more precise CIS device, and defects in Al_2_O_3_ must be further cured. In particular, for application as a dielectric film, the interface trap density (D_it_) should be reduced to increase the amount of light absorbed. There are several causes of trap generation in the interface area between Al_2_O_3_ dielectric and substrate. If the Al_2_O_3_ dielectric is deposited on the silicon substrate, the hydroxyl group (-OH) and Si are bonded, which may act as an interface trap [8]. In another case, carbon impurities generated during the Al_2_O_3_ deposition process act as interface traps.

Carbon impurities were generated after the Al_2_O_3_ film was deposited via plasma-enhanced atomic layer deposition (PEALD) using trimethylaluminum as a precursor [9]. These impurities act as traps inside the Al_2_O_3_ and in the interface region. Previously, residual carbon was removed using the H_2_ plasma treatment of an Al_2_O_3_ film [10]. The quality of the dielectric and interface areas increased with a decrease in carbon impurity contents. In addition, posttreatments provided sufficient fixed charges for the Al_2_O_3_ dielectric to be used as the passivation layer of the CIS [11,12]. However, a low D_it_ is required for next-generation CIS devices. Although well-known defects, such as oxygen vacancies, have been investigated [13], limited studies have been conducted to reduce residual carbon contents, except by changing the precursor [14].

In this study, the oxygen plasma exposure time was adjusted during Al_2_O_3_ deposition to reduce the residual carbon content. The increased O_2_ plasma exposure time sufficiently decreased the D_it_ of the Al_2_O_3_ gate stack. Consequently, it showed a considerably lower D_it_ compared with that of the sample processed via rapid thermal annealing and H_2_ plasma treatment on Al_2_O_3_, which exhibited the lowest D_it_ in a previous study [10]. In particular, D_it_ was the lowest after post-metallization annealing (PMA) to Al_2_O_3_ samples with increased O_2_ plasma exposure time. In addition, a positive shift in flat-band voltage (ΔV_fb_) was prevented by reducing carbon generation. However, D_it_ increases when H_2_ plasma treatment is performed after Al_2_O_3_ deposition. Plasma damage and residual hydrogen impurities were caused by excessive H_2_ plasma treatment on Al_2_O_3_ dielectric and were validated using the C–V hump occurring in the capacitance vs. voltage (C–V_G_) measurement curve.

## 2. Experimental Materials and Methods

As shown in Figure 1, an Al_2_O_3_ film was deposited on a Si substrate at 275 °C using PEALD. Substrate included moderately doped p-type Si (1–30 Ω·cm, (100)) with a doping concentration of ~1.3 × 10^16^ cm^−3^. Prior to deposition of the Al_2_O_3_ layer, Si substrates were cleaned by dipping in a NH_4_OH:H_2_O_2_:H_2_O mixture (1:1:5 by volume), known as Standard Clean 1 (SC1), for 10 min at 70 °C, followed by dipping in dilute HF (100:1) for 1 min to remove native oxides. For deposition of Al_2_O_3_ dielectric, a commercial 200 mm wafer plasma-enhanced vapor deposition (PECVD; Quros Plus 200) was used. As a precursor, Trimethylaluminum (TMA, Al(CH_3_)_3_) (Up chemical co. Ltd., Pyeongtaek, Gyeonggi-do, Republic of Korea; 99.9999%) was supplied. For sequential surface reactions, O_2_ plasma was supplied with TMA. The O_2_ plasma exposure times were 3 and 7 s. During the deposition, an Al(CH_3_)_3_ container temperature of 25 °C, an Ar purge flow rate of 500 sccm, an O_2_ flow rate of 100 sccm and a chamber pressure of 0.4 mTorr were used. Al electrode with a diameter of 300 µm and an area of 7.06 × 10^4^ µm^2^ was deposited on the Al_2_O_3_ dielectric using an e-beam evaporator. The thickness of the Al_2_O_3_ film was measured using transmission electron microscopy (TEM; JEM-2100F; JEOL KOREA LTD., Seoul, Republic of Korea) and ellipsometry (M-2000; J. A. Woollam Co., Anyang, Gyeonggi-do, Republic of Korea). After Al_2_O_3_ deposition, H_2_ plasma treatment and PMA were performed separately depending on the sample (Table 1). H_2_ plasma treatment was performed with a H_2_ gas flow rate ratio {[H_2_] = ([H_2_] + [Ar])} of 0.89 in a PECVD chamber for 15 min. PMA was performed at 400 °C under a N_2_ gas flow in a furnace for 30 min. The N_2_ gas flow rate {[N_2_] = ([N_2_] + [H_2_])} was 0.95 (gas pump: 100 sccm; pressure: 0.7 atm). Under the N_2_ gas flow, the temperature increased from 25 °C to 400 °C in 1 h and then decreased from 400 °C to 25 °C in 2 h. Secondary ion mass spectrometry (SIMS) measurements were conducted on a circular area with a diameter of 33 µm using the Cs+ software. Selective area diffraction pattern (SADP) analysis was carried out to determine crystallinity of the Al_2_O_3_ film. The capacitance and conductance were measured using a B1520A multifrequency capacitance measurement unit at various frequencies (1 kHz–1 MHz). The leakage current and breakdown field were measured using a Keithley 4200-SCS instrument (Tektronix KOREA, Seoul, Republic of Korea). D_it_ (≈2.5(qA)^−1^($G_{p}/\omega$)_max_) was calculated following the well-known conductance method [15]:(1)$$G_{p}/\omega =C_{OX}^{2}G_{M}\omega^{-1}/\left\{{\left(G_{M}/\omega \right)}^{2}+{\left(C_{OX}-C_{M}\right)}^{2}\right\}$$
where q = 1.6 × 10^19^ C; A is the area of the electrode; ($G_{p}/\omega$)_max_ is the normalized parallel conductance peak; $C_{OX}$ is the capacitance in strong accumulation; $C_{M}$ is the measured capacitance; and $G_{M}$ is the measured conductance.

## 3. Results and Discussion

### 3.1. Post-Metallization Annealing

Al_2_O_3_ was deposited via PEALD using trimethylaluminum as the precursor and O_2_ plasma. A flux of O* radicals reacts with methyl groups and is effused in the CO_X_ (x = 1–2) state [16]. However, residual carbon is generated when a sufficient reaction is not performed and acts as a defect in the inner and interfacial regions of Al_2_O_3_. Accordingly, the O_2_ plasma exposure time was increased to 7 s to ensure a sufficient response.

The TEM image of the as-deposited Al_2_O_3_ film is shown in Figure 2a. An Al_2_O_3_ film with a thickness of 30 nm was deposited on the Si substrate, and Al electrode was deposited on the Al_2_O_3_ dielectric. No interfacial layer (IL) was formed at the interface between Si and Al_2_O_3_. Additionally, based on the SADP in Figure 2a, the as-deposited Al_2_O_3_ is in an amorphous state.

PMA was performed at 400 °C for 30 min after Al_2_O_3_ film deposition. After PMA on the Al_2_O_3_ film, oxygen in the dielectric film diffused toward the Si substrate. Accordingly, Si and oxygen form a bond in the SiO_X_ (x = 1–2) state, thereby forming an IL with a thickness of 2.5 nm [8,17] (Figure 2b). As IL was formed between Al_2_O_3_ and Si, the thickness of Al_2_O_3_ decreased from 28.7 to 26.9 nm after PMA. Furthermore, as shown in SADP, amorphous Al_2_O_3_ is converted to polycrystalline Al_2_O_3_ via PMA [18].

The normalized capacitance vs. voltage curves before and after PMA of S1 and S2 are shown in Figure 3. The graphical ((C_OX_/C_MOS_)^2^ − 1)(V_G_) method [19] was applied to the normalized capacitance vs. voltage curve to extract V_fb_. The V_fb_ of as-deposited S1 was 1.65 V, showing a considerable flat-band voltage shift (ΔV_fb_ ≈ 2.54 V) compared with the theoretical value of Al_2_O_3_ dielectric (V_fb_ ≈ −0.89 V). This V_fb_ shift resulted from defects, such as carbon impurities that occur during Al_2_O_3_ deposition via PEALD. However, in the case of S2 samples with an increased O_2_ plasma exposure time, the V_fb_ of S2_as_dep is 0.61 V, exhibiting a smaller ΔV_fb_ compared with that of S1. This is because the amount of negatively charged defects inside S2 is smaller than that of S1.

V_fb_ increased by 0.54 V after PMA in the case of Al_2_O_3_ samples with short O_2_ plasma exposure times. Internal defects that form bonds with carbon impurities have a negative charge and diffuse toward Si [10,14]. However, in the case of S2 samples with long O_2_ plasma exposure times, the change in V_fb_ was as small as 0.2 V owing to a decrease in the defects that can be diffused.

The permittivity of Al_2_O_3_ samples before and after PMA is shown in Figure 4. The permittivity is 9.5 in the case of the as-deposited S1 sample, which is similar to the generally known permittivity value of amorphous Al_2_O_3_ (6–9) [18,20]. However, an IL of SiO_X_ (x = 1–2) is formed between Al_2_O_3_ and Si after PMA, slightly decreasing the permittivity. The permittivity of the as-deposited S2 sample is 12.5, which is considerably higher than that of the S1 sample. This is because of the decrease in the content of various defects and the increase in the internal carbon concentration owing to the longer O_2_ plasma exposure time. After PMA on the as-deposited S2 sample, the permittivity decreases to 10.5 because an IL of SiOx (x = 1–2) is formed between Al_2_O_3_ and Si like the S1 sample. However, the S2_PMA sample still showed a higher permittivity than the S1 samples with shorter O_2_ plasma exposure time.

The decrease in the carbon impurity content with increasing O_2_ plasma exposure time was validated using SIMS depth profiling. As shown in Figure 5, the amount of carbon impurities in the Al_2_O_3_ film deposited with an O_2_ plasma exposure time of 7 s is considerably less than that of the Al_2_O_3_ sample deposited with a shorter O_2_ plasma exposure time. As the O_2_ plasma exposure time increased, more carbon was effused into the CO_X_ (x = 1–2) gas state through numerous reactions between the oxygen plasma and carbon [16]. If the O_2_ plasma exposure time is more than 7 s, there is a possibility of improvement as much as carbon is reduced. However, there is a limit to effuse through the reaction with carbon, and the improvement effect is expected to be saturated as carbon is reduced.

To apply Al_2_O_3_ as a passivation dielectric film, the quality of the interface region between Si and Al_2_O_3_ is crucial. Carbon in Al_2_O_3_ acts as an interface trap in the interface region between the Al_2_O_3_ dielectric and Si substrate [21]. The parallel conductance versus frequency plots of the Al_2_O_3_ films with various D_it_ values are shown in Figure 6. D_it_ was measured using the conductance method [13]. The D_it_ of the S1_as_dep sample was 8.98 × 10^13^ eV^−1^·cm^−2^, whereas that of the S2_as_dep sample was 1.12 × 10^12^ eV^−1^·cm^−2^. The interface traps of the S2 sample decreased with a decrease in the carbon impurity content in the interface area with increasing O_2_ plasma exposure time. After PMA, the interface region between the Al_2_O_3_ dielectric and Si was improved due to various reasons. First, an IL was formed after the application of PMA to the Al_2_O_3_ gate stack. Therefore, the number of hydroxyl groups is reduced, thereby decreasing the number of interface traps [22]. For other reason, as crystallization of Al_2_O_3_ occurred due to PMA, defects and dangling bonds acting as traps in the interface region were removed. In addition, crystallization of the Al_2_O_3_ dielectric stabilized the bond between the Al_2_O_3_ and Si substrate [10].

In summary, the number of interface traps of the S2_PMA sample, in which the concentrations of both carbon impurities and hydroxyl groups were reduced, were the lowest in this study (D_it_ = 1.35 × 10^11^ eV^−1^·cm^−2^).

The interface improvement owing to the increase in the O_2_ plasma exposure time was also validated using the breakdown characteristics. The gate leakage current with an increase in the electrical field of the S1 and S2 Al_2_O_3_ samples is shown in Figure 7a. In the case of S1_as_dep, the breakdown occurred at 9.73 MV/cm. The breakdown characteristics improved after PMA was performed owing to the formation of an IL, which occurred at 11.47 MV/cm. However, breakdown did not occur until the application of the maximum electric field (14 MV/cm) of the 4200-SCS equipment in the case of the S2 sample. Furthermore, breakdown did not occur in the case of the S2_as_dep sample without the IL. This was because of the reduction in the impurity content in the interface area with an increase in the O_2_ plasma exposure time.

In addition, the FN plots to validate the improvement in the interface quality are shown in Figure 7b. The FN plot is analyzed using the leakage current density caused by FN tunneling, *J_FN_*, and can be described as follows:(2)$$J_{FN}=AE^{2}\exp\left(-B/E\right),$$
where $$A=q^{3}m_{0}/\left(8\pi hm^{*}\Phi_{B}\right),$$
and $$B=4{\left(2m^{*}\right)}^{\frac{1}{2}}{\left(q\Phi_{B}\right)}^{\frac{3}{2}}/\left(3qh/2\pi \right),$$
where A is the Richardson’s constant; q is the electronic charge; h is Planck’s constant; $m_{0}$ is the free electron mass; $m^{*}$ is the effective electron mass in the oxide; and $\Phi_{B}$ is the barrier height [23]. The steeper the slope in the FN plot, the larger the FN barrier height $\Phi_{B}$ [4]. Since the absolute value of the slope of the S2_as_dep sample (slope = −182.06) is larger than that of the S1_as_dep sample (slope = −103.28), it means that the barrier height is higher in S2_as_dep. Therefore, the FN plot shows that the interface region of Al_2_O_3_/Si was improved in the S2 sample with increased O_2_ plasma exposure time.

In summary, the increase in the O_2_ plasma exposure time decreases the carbon content in Al_2_O_3_, which reduces D_it_, improves the breakdown field, and prevents the V_fb_ shift. However, the H_2_ plasma treatment decreased the quality of the oxide and interface owing to the increase in the O_2_ plasma exposure time, which is discussed later.

### 3.2. H_2_ Plasma Treatment

H_2_ plasma treatment significantly decreased the carbon impurity content in Al_2_O_3_ in previous studies [10], thereby preventing the V_fb_ shift and improving the breakdown characteristics. However, further improvements in the interface quality is required for next-generation CIS devices. Therefore, we analyzed the effects of the H_2_ plasma treatment on Al_2_O_3_ films with increasing O_2_ plasma exposure time.

D_it_ values depending on various treatments on the Al_2_O_3_ samples are shown in Figure 8. The average D_it_ of the sample with the H_2_ plasma treatment was 4.45 × 10^12^ eV^−1^·cm^−2^, which was significantly smaller than that of as-deposited S1. However, the average D_it_ of the sample with the H_2_ plasma treatment was 1.13 × 10^12^ eV^−1^·cm^−2^ in the case of S2 samples with an increased O_2_ plasma exposure time, which increased compared with the average D_it_ of as-deposited S2 (D_it,as_dep S2_ = 5.79 × 10^11^ eV^−1^·cm^−2^). A similar trend was observed after PMA. D_it_ was higher in the S2 sample with the H_2_ plasma treatment and PMA than that of the S2 sample treated with only PMA.

A large amount of carbon impurities was removed owing to the increased O_2_ plasma exposure time in the S2 sample. Therefore, there are not enough carbon impurities for the reaction with the H_2_ plasma. As a result, owing to the excessive postprocessing H_2_ plasma treatment on the S2 sample, H impurities remained inside the Al_2_O_3_ film [24]. In addition, additional H_2_ plasma treatment for carbon impurities, whose content was reduced owing to an increase in the O_2_ plasma exposure time, had a more significant effect on the formation of defects owing to damage due to the plasma treatment compared with the effects of curing defects owing to carbon content reduction [25]. In conclusion, in the case of the S2 sample with increased O_2_ plasma exposure time, excessive postprocessing H_2_ plasma treatment caused residual H impurities and plasma damage, which contributed to increase D_it_ by forming dangling bonds in interface region.

Using the capacitance vs. voltage curve, the plasma damage to the gate stack was confirmed. The normalized capacitance before and after the H_2_ plasma treatment in the S2 sample with an increased O_2_ plasma exposure time is shown in Figure 9. In contrast to the S2_as_dep sample, the C–V hump occurs near V_fb_ in the S2_H_2_ plasma sample. Therefore, the hydrogen plasma, which should be effused via the reaction with carbon, damaged the Al_2_O_3_ dielectric.

The formation of defects in the oxide and interface regions of the S2 sample owing to the H_2_ plasma treatment resulted in more leakage flow in the gate stack. In contrast to the S2_as_dep sample, where breakdown does not occur even under the electric field limit of the 4200-SCS equipment (E_field_ = 14 MV/cm), the breakdown occurs at 11.2 MV/cm in the S2_H_2_ plasma sample (Figure 10). As a result, in the case of the Al_2_O_3_ film with increased O_2_ plasma exposure time, H_2_ plasma treatment rather deteriorates the interface quality between Al_2_O_3_ dielectric and Si.

In summary, H_2_ plasma treatment has different effects depending on the O_2_ plasma exposure time during deposition of the Al_2_O_3_ dielectric. H_2_ plasma treatment was effective for S1 samples with a large amount of carbon impurities because of the short O_2_ plasma exposure time. Due to the reduction of carbon impurities, the D_it_ of the S1 sample was greatly reduced after H_2_ plasma treatment. However, the treatment effects on S2 samples was rather poor, resulting in reduced carbon content owing to the long O_2_ plasma exposure time. H_2_ plasma treatment produced residual H impurities in the S2 samples and also caused plasma damage. Therefore, H_2_ plasma treatment rather increased D_it_ in the Al_2_O_3_ with increased O_2_ plasma exposure time.

## 4. Conclusions

The criterion of fixed charges in the Al_2_O_3_ film for application as a CIS passivation layer was satisfied in a previous study; however, the issue of interface traps remained unresolved. Further improvement in the interface area is required for Al_2_O_3_ to be used as a passivation dielectric layer. Therefore, this study investigated the conditions to reduce defect contents and the D_it_ of the Al_2_O_3_ film. The carbon content inside the Al_2_O_3_ was significantly decreased by adjusting the O_2_ plasma exposure time to induce more reactions during dielectric deposition. D_it_ was significantly decreased owing to the reduction in the amount of carbon impurities, and the improvement in the interface region was validated using the breakdown characteristics. Moreover, H_2_ plasma treatment effectively reduced D_it_ in Al_2_O_3_ films with a short O_2_ plasma exposure time during deposition. However, H_2_ plasma treatment of the Al_2_O_3_ film deposited with a long O_2_ plasma exposure time rather increased D_it_ due to plasma damage. PMA slightly decreased the permittivity after Al_2_O_3_ deposition; however, D_it_ significantly decreased. In particular, in the case of Al_2_O_3_ samples with increased O_2_ plasma exposure time, after PMA, it had the lowest D_it_, which is suitable for use as a passivation layer for CIS.

## Figures and Tables

**Figure 1 nanomaterials-13-00731-f001:**
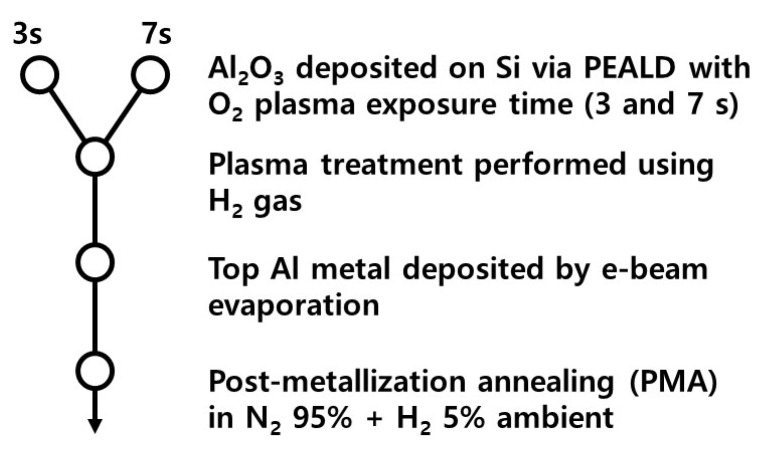
Process flow for the fabrication of the Al/Al_2_O_3_/Si gate stack.

**Figure 2 nanomaterials-13-00731-f002:**
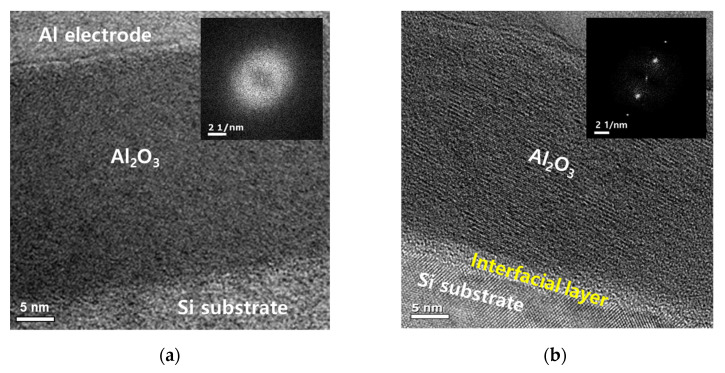
Transmission electron microscope (TEM) image and selective area diffraction pattern (SADP) of (**a**) as-deposited Al/Al_2_O_3_/Si gate stack and (**b**) Al/Al_2_O_3_/Si gate stack after PMA at 400 °C under a N_2_ gas flow in a furnace for 30 min.

**Figure 3 nanomaterials-13-00731-f003:**
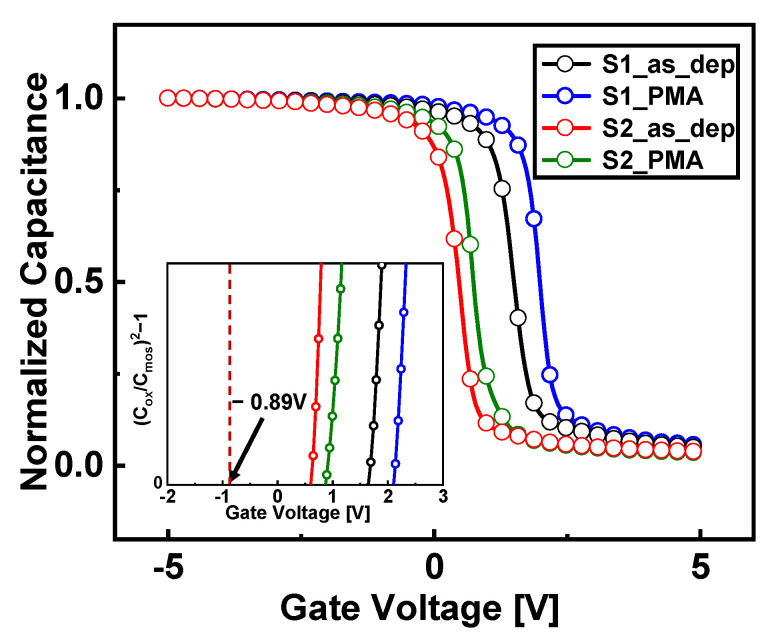
Normalized capacitance vs. voltage graph and graphical ((C_OX_/C_MOS_)^2^ − 1)(V_G_) method to extract the flat-band voltage (V_fb_) of Al_2_O_3_ samples with and without PMA (frequency = 1 MHz).

**Figure 4 nanomaterials-13-00731-f004:**
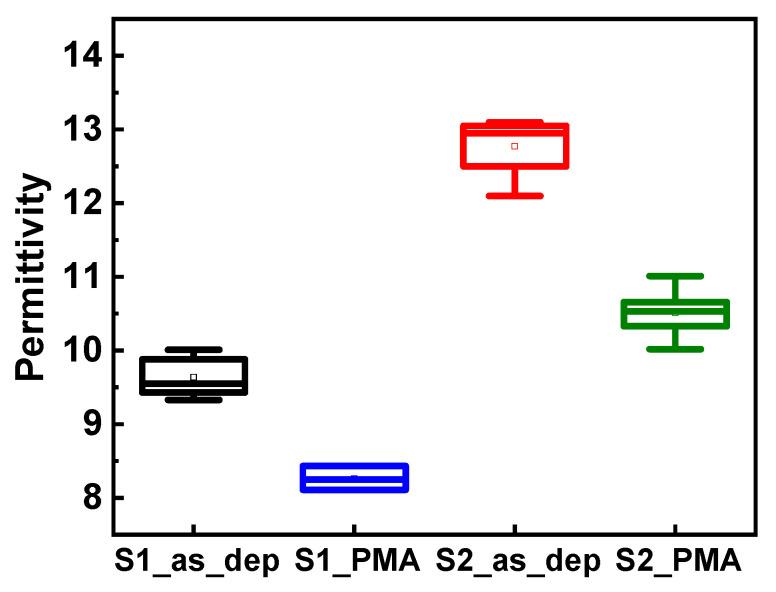
Permittivity of Al_2_O_3_ samples under deposition conditions (O_2_ plasma exposure time: 3 and 7 s) and post-metallization annealing (PMA).

**Figure 5 nanomaterials-13-00731-f005:**
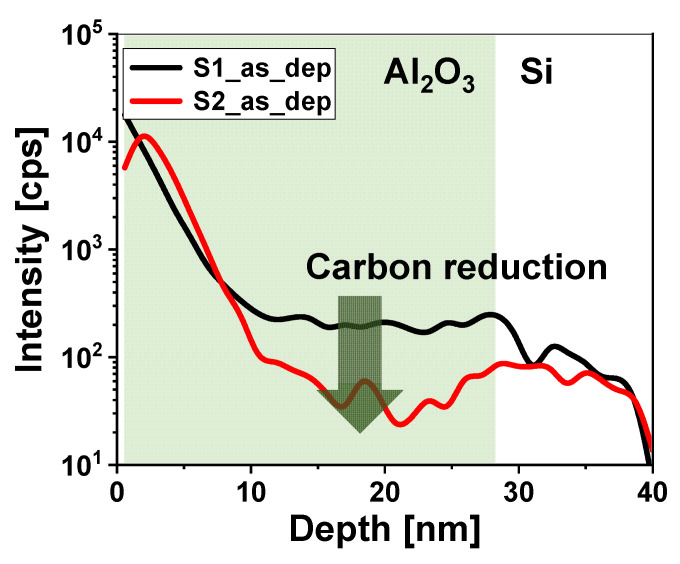
Secondary ion mass spectrometry (SIMS) depth profiles of carbon in the Al/Al_2_O_3_/Si gate stack with an O_2_ plasma exposure time of 3 (black line) and 7 s (red line).

**Figure 6 nanomaterials-13-00731-f006:**
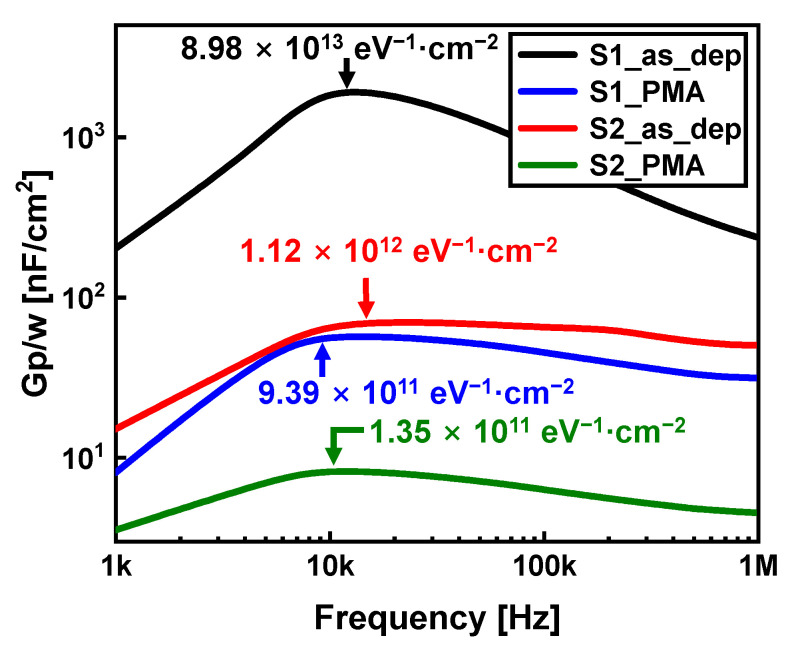
G_P_/w vs. frequency curves of Al_2_O_3_ samples with and without PMA for measuring interface trap density (D_it_).

**Figure 7 nanomaterials-13-00731-f007:**
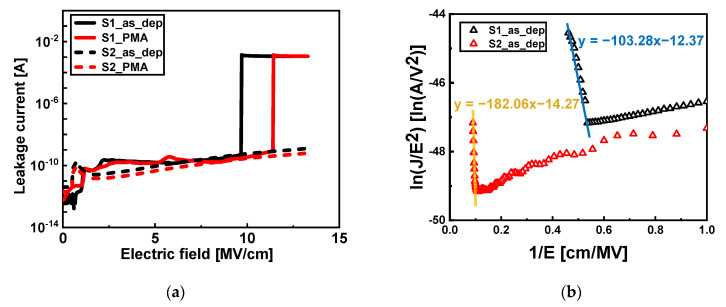
(**a**) Leakage current vs. gate electric field of Al_2_O_3_ samples with and without PMA. (**b**) Fowler–Nordheim (FN) plots of I–V curves for as-deposited Al_2_O_3_ samples with an O_2_ plasma exposure time of 3 (black triangles) and 7 s (red triangles).

**Figure 8 nanomaterials-13-00731-f008:**
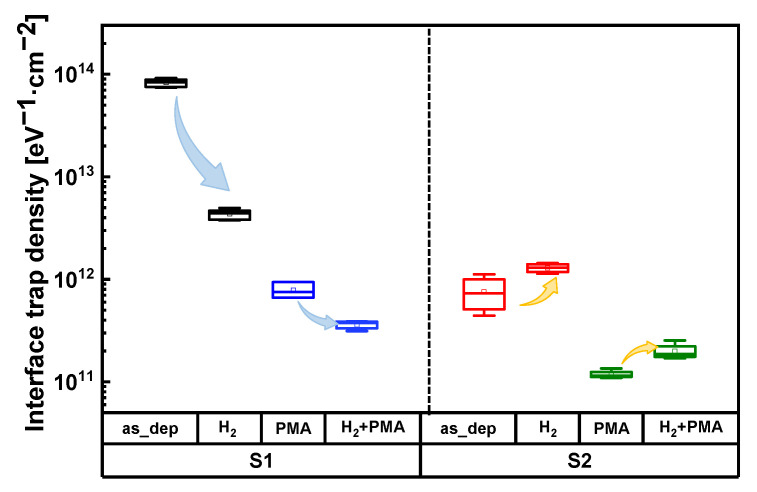
Interface trap density (D_it_) of Al_2_O_3_ samples under deposition conditions; O_2_ plasma exposure time: 3 (**left** side) and 7 s (**right** side) and posttreatment conditions: H_2_ plasma treatment and post-metallization annealing (PMA).

**Figure 9 nanomaterials-13-00731-f009:**
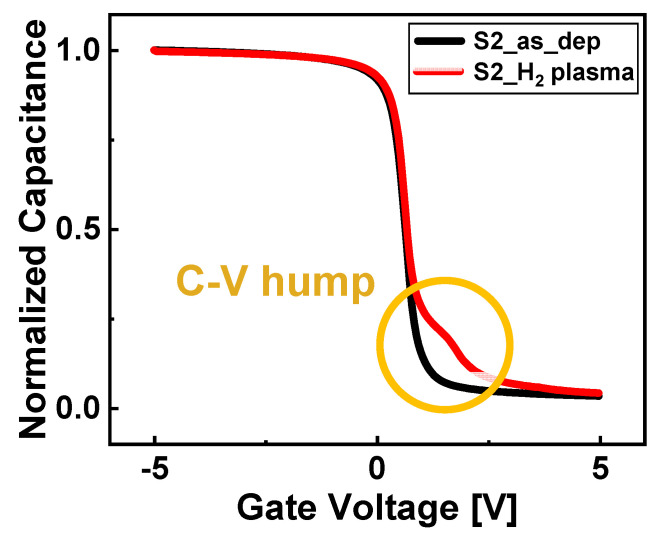
C–V hump effect (yellow circle) owing to excessive H_2_ plasma treatment in the normalized capacitance vs. voltage curves of Al_2_O_3_ films with an increased O_2_ plasma exposure time (frequency = 1 MHz).

**Figure 10 nanomaterials-13-00731-f010:**
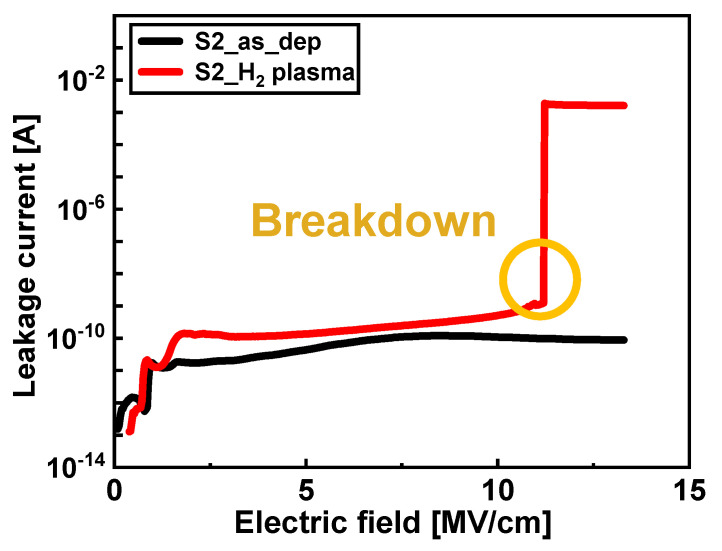
Leakage current vs. gate electric field of Al_2_O_3_ samples with an increased O_2_ plasma exposure time.

**Table 1 nanomaterials-13-00731-t001:** Al_2_O_3_ samples under deposition conditions (O_2_ plasma exposure time: 3 and 7 s) and posttreatment conditions (H_2_ plasma treatment and post-metallization annealing (PMA)).

Samples	O_2_ Plasma Exposure Time (s)	H_2_ Plasma Treatment	PMA
S1_as_dep	3	X	X
S1_H_2__plamsa	3	O	X
S1_PMA	3	X	400 °C; 30 min
S1_H_2_ plasma + PMA	3	O	400 °C; 30 min
S2_as_dep	7	X	X
S2_H_2__plamsa	7	O	X
S2_PMA	7	X	400 °C; 30 min
S2_H_2_ plasma + PMA	7	O	400 °C; 30 min

## Data Availability

Not applicable.

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
