# Peer review of "H_2_ Plasma and PMA Effects on PEALD-Al_2_O_3_ Films with Different O_2_ Plasma Exposure Times for CIS Passivation Layers"

_nanomaterials, 2023, doi:10.3390/nano13040731_

Round 1

Reviewer 1 Report

The main aim of the work under review is to improve the electrical properties of Al2O3 films dedicated for passivation layers in CMOS image sensors by applying the appropriate O2 exposure time and additional processing in H2 plasma. In the paper, the authors indicated the clear trend concerning the studied parameters in order to reduce defect content and the interface trap density in Al2O3 layers.

The manuscript describes a well-planned experiment that was carried out in a systematic manner. The applied research techniques correspond well to the presented scientific problem. The work is rather technical in nature and does not present a deep analysis of the processes taking place in the material under study. However, it can be helpful for scientists and engineers working on the fabrication of Al2O3 passivation layers by showing the general trends regarding the electrical parameters of the Al2O3 subjected to plasma treatment and post metallization annealing.

The manuscript is clearly presented and the data are scientifically sound. However, I would suggest considering a few minor amendments and improvements before publishing:

1. Please add information on how the Si substrate was treated before deposition of the Al2O3 layer. Has the native SiOx layer been removed?

2. The conductivity of silicon given by the authors suggests that the substrate was doped. Please specify what dopant and in what concentration has been added to silicon.

3. Please provide additional parameters of the Al2O3 deposition process, such as precursor concentration and pressure in a preparation chamber.

4. Please provide information on the area of the prepared samples. What area was the electrical signal collected from?

5. Please discuss in the text of the paper what value of interface trap density is required in Al2O3 for application as a passivation layer in CMOS image sensors and how far the described methods are from this goal.

6. According to the authors, can further increase of O2 plasma exposure time cause even greater reduction of carbon content and further improvement of the quality of the Al2O3 layer? Please discuss such possibility in the text of the paper.

7. Please enlarge the scale bar in inset of Fig. 2a and add a scale bar to inset in Fig. 2b.

Author Response

Response to Reviewer 1 Comments

We thank you for the valuable suggestions, which have helped improve our manuscript's quality. We have thoroughly revised the manuscript, taking into account all comments and we have no disagreement with any comments. To meet the total word counts, we put in more information. A summary of our responses and revisions is given below for your convenience.

Black: reviewer comments

Red: our response

Blue: modified contents in the manuscript

Points1: Please add information on how the Si substrate was treated before deposition of the Al2O3 layer. Has the native SiOx layer been removed?

Response1: Thank you for your suggestion. As the reviewer suggested, we added information about treatment of the Si substrate. Prior to deposition of the Al2O3 layer, the native SiOx layer was removed by cleaning with NH4OH : H2O2 : H2O mixture (1:1:5 by volume) and dilute HF (100 : 1).

Revised contents 1:

- On page 3, Experimental Materials and Methods:

Prior to deposition of Al2O3 layer, Si substrates were cleaned by dipping in a NH4OH : H2O2 : H2O mixture (1:1:5 by volume), known as Standard Clean 1 (SC1), for 10 min at 70 °C, followed by dipping in dilute HF (100 : 1) for 1 min to remove native oxides.

Points2: The conductivity of silicon given by the authors suggests that the substrate was doped. Please specify what dopant and in what concentration has been added to silicon.

Response2: We thank you for your comments. Doping concentration of the p-type Si substrate is about 1.3 x 1016 cm-3. As the reviewer suggested, the dopant type and doping concentration were added to the manuscript.

Revised contents 2:

- On page 3, Experimental Materials and Methods:

Substrate included moderately doped p-type Si [1–30 Ω·cm, (100)] with a doping concentration of ~ 1.3 x 1016 cm-3.

Points3: Please provide additional parameters of the Al2O3 deposition process, such as precursor concentration and pressure in a preparation chamber.

Response3: We thank you for your comments. As a precursor, Trimethylaluminum [TMA, Al(CH3)3] (Up chemical co. Ltd., 99.9999 %) was supplied. During the Al2O3 deposition process, an Al(CH3)3 container temperature of 25 °C, an Ar purge flow rate of 500 sccm, an O2 flow rate of 100 sccm, a chamber pressure of 0.4 mTorr, and a substrate temperature of 300 °C were used. We added the detailed Al2O3 deposition process in the manuscript.

Revised contents 3:

- On page 3, Experimental Materials and Methods:

For deposition of Al2O3 dielectric, a commercial 200 mm wafer plasma-enhanced vapor deposition (PECVD; Quros Plus 200) was used. As a precursor, Trimethylaluminum [TMA, Al(CH3)3] (Up chemical co. Ltd., 99.9999 %) was supplied. For sequential surface reactions, O2 plasma was supplied with TMA. The O2 plasma exposure times were 3 and 7 s. During the deposition, an Al(CH3)3 container temperature of 25 °C, an Ar purge flow rate of 500 sccm, an O2 flow rate of 100 sccm and a chamber pressure of 0.4 mTorr were used.

Points4: Please provide information on the area of the prepared samples. What area was the electrical signal collected from?

Response4: Thank you for your comments. Al electrode for electric signal is deposited on the Al2O3 dielectric. Al electrode is circular, with a diameter of 300 µm and an area of 7.06 x 104 µm2. The information about electrode was added to the manuscript.

Revised contents 4:

- On page 3, Experimental Materials and Methods:

Al electrode with a diameter of 300 µm and a circular area of 7.06 x 104 µm2 was deposited on the Al2O3 dielectric using an e-beam evaporator.

Points5: Please discuss in the text of the paper what value of interface trap density is required in Al2O3 for application as a passivation layer in CMOS image sensors and how far the described methods are from this goal.

Response5: We thank you for your valuable comments. Our target for using the Al2O3 layer as a passivation dielectric film for CIS is to contain a Dit of 1.0 × 1011 eV-1·cm-2 or less. There has been a study that a Dit of about 1.0 × 1011 eV-1·cm-2 was obtained by using a SiO2 for CIS, which is not a high-k material [R1]. Thus, we thought that there would be no problem as a CIS passivation layer if Dit was about 1.0 × 1011 eV-1·cm-2.

We significantly lowered Dit to 1.35 × 1011 eV-1·cm-2 by increasing the O2 plasma exposure time and by PMA at the 400 °C. We have almost reached our goal. We revised the manuscript to explain the outcome of this study.

Revised contents 5:

- On page 11, Conclusions:

In particular, in the case of Al2O3 samples with increased O2 plasma exposure time, after PMA, it had the lowest Dit, which is suitable for use as a passivation layer for CIS.

Points6: According to the authors, can further increase of O2 plasma exposure time cause even greater reduction of carbon content and further improvement of the quality of the Al2O3 layer? Please discuss such possibility in the text of the paper.

Response6: We appreciate the reviewer’s insightful comments. If the O2 plasma exposure time is increased to more than 7 s, there is a possibility of improvement as much as carbon is reduced, but there is a limit to the reaction with carbon, and the improvement effect is expected to be saturated as carbon is reduced.

Revised contents 6:

On page 6, 2nd paragraph:

If the O2 plasma exposure time is more than 7 s, there is a possibility of improvement as much as carbon is reduced. However, there is a limit to effuse through the reaction with carbon, and the improvement effect is expected to be saturated as carbon is reduced.

Points7: Please enlarge the scale bar in inset of Fig. 2a and add a scale bar to inset in Fig. 2b.

Response7: Thank you for your detailed comments. As the reviewer suggested, we increased the size of the SADP in both Fig. 2a and Fig. 2b to better show the diffraction pattern as well as the scale bar. In addition, we added a scale bar of SADP in Fig. 2b.

Revised contents 7:

On page 4. Figure 2:

(a)

(b)

Revised Figure 2. Transmission electron microscope (TEM) image and selective area diffraction pattern (SADP) of (a) as-deposited Al/Al2O3/Si gate stack and (b) Al/Al2O3/Si gate stack after PMA at 400 °C under a N2 gas flow in a furnace for 30 min.

References

[R1] Onaka-Masada, A.; Kadono, T.; Okuyama, R.; Hirose, R.; Kobayashi, K.; Suzuki, A.; Koga, Y.; Kurita, K. Reduction of Dark Current in Cmos Image Sensor Pixels Using Hydrocarbon-Molecular-Ion-Implanted Double Epitaxial Si Wafers. Sensors (Switzerland) 2020, 20, 1–18, doi:10.3390/s20226620.

Reviewer 2 Report

Dear Authors,

the paper is well organized still it 

requires some corrections to be performed.

1)

In lines 172-178, authors describe the barrier height,

but they base on a declared electron effective mass.

Since there is no measure of this value in the specific prepared layer

it is better to give only the extracted Richardson constants for S1 and S2.

2)

Figure 10 shows breakdown around 11 MV/cm.  Line 223 states 12.1

Is it a misprint for 11.2 or graph scale is wrong ?

3) Conclusions are not well ordered. H2 treatment at the end is not useful

since best results are in S2-PMA according to me. Line 239-242 should be moved upward of lines 233-239 or directly eliminated.

Author Response

Response to Reviewer 2 Comments

We thank you for the valuable suggestions, which have helped improve our manuscript's quality. We have thoroughly revised the manuscript, taking into account all comments and we have no disagreement with any comments. To meet the total word counts, we put in more information. A summary of our responses and revisions is given below for your convenience.

Black: reviewer comments

Red: our response

Blue: modified contents in the manuscript

Points1: In lines 172-178, authors describe the barrier height,

but they base on a declared electron effective mass.

Since there is no measure of this value in the specific prepared layer

it is better to give only the extracted Richardson constants for S1 and S2.

Response1: We appreciate the reviewer’s valuable comments to improve our manuscript. As the reviewer suggested, we removed the barrier height value using electron effective mass from the manuscript and revised the Figure 7. Since the Richardson constants used in the FN plot are obtained by intercept, the value changes too much even with a slight error. Therefore, we wrote the linear fit function including the slope, which is one of the ways to infer the barrier height, as well as the Richardson constants to compare the barrier height and interface characteristics of Al2O3 samples.

Revised contents 1:

- On page 7, Figure 7:

(a)

(b)

Revised Figure 7. (a) Leakage current vs. gate electric field of Al2O3 samples with and without PMA. (b) Fowler-Nordheim (FN) plots of I-V curves for as-deposited Al2O3 samples with an O2 plasma exposure time of 3 (black triangles) and 7 s (red triangles).

- On page 8, 2nd paragraph:

The steeper the slope in the FN plot, the larger the FN barrier height  [3]. Since the absolute value of the slope of the S2_as_dep sample (slope = -182.06) is larger than that of the S1_as_dep sample (slope = -103.28), it means that the barrier height is higher in S2_as_dep. Therefore, the FN plot shows that the interface region of Al2O3/Si was improved in the S2 sample with increased O2 plasma exposure time.

Points2: Figure 10 shows breakdown around 11 MV/cm.  Line 223 states 12.1

Is it a misprint for 11.2 or graph scale is wrong ?

Response2: We appreciate for your detailed check. By mistake, we wrote 12.1 MV/cm in the text, which was the average of several measurements, not the value used in the graph in Figure 10. In the graph of Figure 10, the value of the electrical field where breakdown occurred is 11.26 MV/cm. Thus, we revised the manuscript. Even if the breakdown field is changed from 12.1 MV/cm to 11.2 MV/cm, there is no problem with the overall data interpretation and trend.

Revised contents 2:

- On page 10, 2nd paragraph:

In contrast to the S2_as_dep sample, where breakdown does not occur even under the electric field limit of the 4200-SCS equipment (Efield = 14 MV/cm), the breakdown occurs at 11.2 MV/cm in the S2_H2 plasma sample (Fig. 10).

Points3: Conclusions are not well ordered. H2 treatment at the end is not useful

since best results are in S2-PMA according to me. Line 239-242 should be moved upward of lines 233-239 or directly eliminated.

Response3: Thank you for your detailed comments. We wrote the conclusion by following the experimental sequence. However, as the reviewer suggested, the effect of H2 plasma treatment was a minor part, and the effect of PMA was important. Thus, we revised the conclusion of the manuscript. The content of H2 plasma treatment was moved upward, and the content that S2-PMA was the best result was added at the end of the conclusion.

Revised contents 3:

- On page 11, Conclusions:

Also, H2 plasma treatment effectively reduced Dit in Al2O3 films with a short O2 plasma exposure time during deposition. However, H2 plasma treatment of the Al2O3 film deposited with a long O2 plasma exposure time rather increased Dit due to plasma damage.

- On page 11, Conclusions:

In particular, in the case of Al2O3 samples with increased O2 plasma exposure time, after PMA, it had the lowest Dit, which is suitable for use as a passivation layer for CIS.
